# Formation Features of Hybrid Nanocomposites Based on Polydiphenylamine-2-Carboxylic Acid and Single-Walled Carbon Nanotubes

**DOI:** 10.3390/polym11071181

**Published:** 2019-07-13

**Authors:** Sveta Zhiraslanovna Ozkan, Galina Petrovna Karpacheva, Aleksandr Ivanovich Kostev, Galina Nikolaevna Bondarenko

**Affiliations:** A.V. Topchiev Institute of Petrochemical Synthesis, Russian Academy of Sciences, 29 Leninsky prospect, 119991 Moscow, Russia

**Keywords:** polydiphenylamine-2-carboxylic acid, single-walled carbon nanotubes, conjugated polymers, in situ oxidative polymerization, hybrid nanocomposites

## Abstract

Hybrid nanocomposites based on electroactive polydiphenylamine-2-carboxylic acid (PDPAC) and single-walled carbon nanotubes (SWCNTs) were obtained for the first time. Polymer-carbon nanomaterials were synthesized via in situ oxidative polymerization of diphenylamine-2-carboxylic acid (DPAC) in the presence of SWCNTs by two different ways. Hybrid SWCNT/PDPAC nanocomposites were prepared both in an acidic medium and in the heterophase system in an alkaline medium. In the heterophase system, the monomer and the SWCNTs are in the organic phase (chloroform) and the oxidant (ammonium persulfate) is in an aqueous solution of ammonium hydroxide. The chemical structure, as well as the electrical and thermal properties of the developed SWCNT/PDPAC nanocomposite materials were investigated.

## 1. Introduction

One of the most promising areas of development in the nanotechnology industry is the creation of nanomaterials that offer properties required by modern technologies. Researchers are particularly interested in hybrid nanocomposites that include carbon nanotubes (CNTs) [1,2,3,4,5] dispersed in a conjugated polymer matrix [6,7,8,9]. Conjugated polymers are a special class of polymer materials, the characteristic feature of which is the delocalization of π-electrons along the conjugation chain [10,11,12]. The specific electronic structure of polymers with a system of conjugated double bonds determines the electrophysical, electrochemical, optical, and other properties of these materials [13,14]. Due to the electronic interaction of polymer and carbon constituents in such systems, fundamentally new or enhanced properties, as compared to initial components, can be expected to appear (thermal stability, mechanical strength, electrical and thermal conductivity, etc.). This increases the potential range of their practical application and opens opportunities for solving numerous technological problems in electronics, microsystem technology, energy industry, engineering, medicine, etc. Due to the complementary properties, nanocomposite materials based on CNTs and conjugated polymers are promising for use in organic electronics, health care, for creating optoelectronic devices, thin film transistors, memory modules, electrochemical power sources, supercapacitors, sensors, displays, etc.

Nowadays, many methods for the production of nanocomposites based on CNTs and conjugated polymers have been developed. Polyaniline (PANI), polypyrrole, and polythiophene, as well as their derivatives are used as polymer components [6,7,8,9]. A serious problem in obtaining these nanocomposites is the tendency of CNTs to aggregate. The CNTs’ aggregation prevents their homogeneous distribution in a polymer matrix and therefore does not allow the required properties to be obtained. The cause for this lies in the insolubility of CNTs in organic solvents and the lack of compatibility of the polymer with CNTs. The use of electrodeposition of polymers onto the CNT surface [15,16,17,18,19,20], electrochemical polycondensation of monomers on the surface of oriented CNTs [21], and dissolving polymers in a CNT suspension in an organic solvent [22,23] to produce hybrid materials does not allow a high dispersion of the carbon component in the polymer matrix.

In situ oxidative polymerization in the presence of CNTs is an effective method for the prevention of CNT aggregation [24,25,26,27,28,29,30]. The use of an ultrasound ensures a uniform CNT distribution in the reaction medium and prevents the nanotubes’ coagulation during polymerization [31,32,33]. Studies of oxidative polymerization of aniline in the presence of CNTs showed that PANI forms a uniform polymer coating on the surface of carbon nanomaterials [34,35,36,37,38]. The formation of polymer chains on the CNT surface causes a strong π–π* interaction [23,36,39,40] between the nanomaterial components. The π–π* interaction occurs due to the charge transfer from the quinoid units of PANI to the aromatic structures of CNTs. This enhances the electron transport in the nanocomposite material compared to the pure polymer, thus improving its electrical properties [25,31,32]. The electrical conductivity of MWCNT/PANI increased with MWCNT loading, especially at low loadings, typically below 5 wt %. It was attributed to the induced crystallinity of PANI at the MWCNT surface [25].

Also, high dispersion of the CNT distribution can be achieved by CNT surface modification due to the covalent bonding of functional groups [41,42,43,44,45]. In situ polymerization in the presence of covalently functionalized CNTs not only provides the formation of a homogeneous nanostructure, but also prevents phase micro-separation. The PANI chains deposited on the surface of carboxylic acid functionalized multi-walled carbon nanotubes (MWCNTs) have longer conjugation lengths than the pure PANI [45]. This could be attributed to the site-selective interaction between the conjugated structure of PANI via the quinoid ring and the π-bonded surface of functionalized MWCNTs [36].

A comparative study of MWCNT/PANI with SWCNT/PANI nanocomposite films showed that the electrical conductivity was higher with the use of SWCNTs. This is due to the formation of the core-shell SWCNT-PANI structure and the absence of the agglomerated polymerized PANI islands, which are less conductive. The composite films reached their percolation threshold at the 0.1 wt % MWCNT loading and at the 0.05 wt % SWCNT loading [46].

Earlier, we obtained a hybrid nanomaterial based on poly-3-amine-7-methylamine-2-methylphenazine (PAMMP) and single-walled carbon nanotubes (SWCNTs) [47,48]. The nanocomposite synthesis was carried out in an aqueous solution of acetonitrile via oxidative polymerization of 3-amine-7-dimethylamine-2-methylphenazine hydrochloride in the presence of SWCNTs. A shift of the skeletal oscillation frequencies of PAMMP indicated the π–π* interaction between the phenazine units of the polymer and SWCNTs. 

In this research paper, nanocomposite materials based on polydiphenylamine-2-carboxylic acid (PDPAC) and SWCNTs were prepared for the first time. Hybrid SWCNT/PDPAC nanomaterials were synthesized via in situ oxidative polymerization of diphenylamine-2-carboxylic acid (DPAC) in the presence of SWCNTs in an alkaline and an acidic media. Single-walled carbon nanotubes were chosen because they are characterized by a smaller diameter and better dispersibility in the polymer matrix. Due to this, a greater amount of polymer formed on the CNT surface, forming the core-shell structure [46]. PDPAC is a novel conjugated polyacid synthesized by the authors [49,50,51], which is the N-substituted polyaniline containing an aromatic substituent with a carboxyl group in the *orto* position. The presence of the functional group can be expected to determine the electronic interaction between CNTs and the polymer not only through the main polymer chain but also through side substitutes. The influence of the reaction medium pH and the SWCNT content on the structure, morphology, thermal stability, and electrical properties of nanocomposites was established. The use of PDPAC in the nanocomposites expands the range of conjugated polymers involved in the creation of novel materials for modern technologies.

## 2. Experimental

### 2.1. Materials

Ammonium persulfate (analytical grade) was purified by recrystallization from distilled water by a known procedure [52]. Diphenylamine-2-carboxylic acid (N-phenylanthranilic acid) (C_13_H_11_O_2_N) (analytical grade), aqueous ammonia (reagent grade), sulfuric acid (reagent grade), chloroform (reagent grade), and DMF (Acros Organics) were used as received without any additional purification. The aqueous solutions of reagents were prepared with the use of bidistilled water. SWCNTs from Carbon Chg, Ltd. (Moscow, Russia), with values of *d* = 1.4 to 1.6 nm, and *l* = 0.5 to 1.5 µm, were produced by the electric arc discharge technique with Ni/Y catalyst. 

### 2.2. Preparation of SWCNT/PDPAC Nanocomposites

The SWCNT/PDPAC nanocomposites were prepared by two ways. The synthesis of the nanocomposite in the heterophase system in an alkaline medium (SWCNT/PDPAC-1) was carried out as follows. First, the required amount of monomer (DPAC) (0.1 mol/L, 0.64 g) was dissolved in a mixture of an organic solvent—chloroform (15 mL) and alkali (NH_4_OH) (0.5 mol/L, 2.3 mL). The SWCNTs were added to the resulting solution. The content of carbon nanotubes was C_SWCNT_ = 1 to 3, 10 wt % relative to the monomer weight. The process was carried out at room temperature with constant intensive stirring for 1 h. Then, for the in situ oxidative polymerization of DPAC in the presence of SWCNTs, an aqueous solution (15 mL) of an oxidizing agent (ammonium persulfate) (0.2 mol/L, 1.368 g) was added to the SWCNT/DPAC suspension in a mixture of chloroform and NH_4_OH. Ammonium persulfate was added in one go, without gradual dosing of reagents. The SWCNT/DPAC suspension was pre-cooled to 0 °C by using the LOIP FT-311-25 cryothermostat (Saint-Petersburg, Russia). The volume ratio of organic and aqueous phases was 1:1 (V_total_ = 30 mL). The synthesis was carried out for 3 h at 0 °C under intensive stirring using an electronic stirrer with a RW 16 basic upper drive (Ika Werke, Germany). A narrow cylinder-shaped round-bottom two-neck flask was used for increasing the efficiency of stirring. When the reaction was completed, the mixture was precipitated in a 10-fold excess of a 2% solution of H_2_SO_4_. The resulting product was filtered off, washed repeatedly with distilled water to remove residual amounts of reagents, and vacuum-dried over KOH to constant weight. The yield of the SWCNT/PDPAC-1 nanocomposite was 0.44 g (67.56%) with C_SWCNT_ = 2 wt %.

The synthesis of the nanocomposite in an acidic medium (SWCNT/PDPAC-2) was carried out as follows. First, the SWCNTs were added to the monomer solution (0.1 mol/L, 0.64 g) in 5 M H_2_SO_4_. The SWCNT/DPAC suspension was stirred in an ultrasonic bath (UZV-2414, Vologda, Russia) at room temperature for 0.5 h. The mixture was heated to 25 °C. The content of carbon nanotubes was C_SWCNT_ = 1 to 3, 10 wt % relative to the monomer weight. Then, for the oxidative polymerization of DPAC in the presence of SWCNTs, aqueous solution of ammonium persulfate (0.2 mol/L, 1.368 g) in the same solvent (1/4 of the total volume, V_total_ = 30 mL) was added to the SWCNT/DPAC suspension. The oxidizer was added drop-wise under intensive stirring. The SWCNT/DPAC suspension was pre-cooled to 0 °C. The synthesis continued for 3 h with intense stirring at 0 °C. When the synthesis was completed, the reaction mixture was precipitated in 200 mL of distilled water. The resulting product was filtered off, and washed repeatedly with 1% solution of H_2_SO_4_ to remove residual reagents. The product was vacuum-dried over CaCl_2_ to constant weight. The yield of the SWCNT/PDPAC-2 nanocomposite was 0.57 g (86.47%) at C_SWCNT_ = 3 wt %.

### 2.3. Synthesis of PDPAC

For comparison with nanocomposites, polydiphenylamine-2-carboxylic acid (PDPAC) was prepared under the same conditions. The PDPAC-1 was synthesized via oxidative polymerization in the heterophase system in an alkaline medium (C_monomer_ = 0.1 mol/L, C_oxidizer_ = 0.2 mol/L, C_alkali_ = 0.5 mol/L) [49]. The PDPAC-2 was obtained in the homogeneous acidic medium (C_monomer_ = 0.1 mol/L, C_oxidizer_ = 0.2 mol/L, C_acid_ = 5 mol/L) [50,51].

### 2.4. Characterization

Attenuated total reflection (ATR) FTIR spectra of the samples in the attenuated total reflectance mode were measured by using a HYPERION-2000 IR microscope (Bruker, Karlsruhe, Germany). The microscope was coupled with the Bruker IFS 66v FTIR spectrometer (Karlsruhe, Germany). The optical range was 600 to 4000 cm^−1^ (150 scans, ZnSe crystal, resolution of 2 cm^−1^).

Electronic absorption spectra of samples in DMF were registered by using an UV-1700 spectrophotometer (Shimadzu, Kyoto, Japan) in the range of 190 to 1100 nm.

The ^13^C solid-state CP/MAS NMR spectra were recorded on an Infinity INOVA 500 NMR spectrometer (Varian Inc., Palo Alto, California, USA). CP/MAS NMR spectra for ^13^C nuclei were registered by using the direct polarization (90° pulse at ^13^C was 2.5 μs, time lapse between scans was 30 s) [53]. Crystalline adamantane was used as the secondary external standard of the chemical shift scale [54,55].

An electron microscopic study was performed by using a Zeiss Supra 25 FE-SEM field emission scanning electron microscope (Carl Zeiss AG, Jena, Germany). 

An X-ray diffraction study was performed in ambient atmosphere by using a Difray-401 X-ray diffractometer (Scientific Instruments Joint Stock Company, Saint-Petersburg, Russia) with Bragg-Brentano focusing on Cr*K*_α_ radiation with *λ* = 0.229 nm. 

Thermogravimetric analysis (TGA) was carried out with a Mettler Toledo TGA/DSC1 (Columbus, Ohio, USA) in the dynamic mode. The test temperature ranged from 30 to 1000 °C (100 mg sample, 10 °C/min heating rate, air and argon atmosphere, 10 mL/min argon flow velocity). The samples were analyzed in an Al_2_O_3_ crucible.

Differential scanning calorimetry (DSC) was performed on a Mettler Toledo DSC823^e^ calorimeter (Columbus, Ohio, USA). The investigation was carried out from room temperature to 350 °C at a heating rate of 10 °C/min under the nitrogen flow of 70 mL/min.

A 6367A precision LCR meter (Microtest Co., Ltd., New Taipei City, Taiwan) was used to measure the *ac* conductivity in the range of frequency of 0.1 Hz to 1.15 MHz.

## 3. Results and Discussion

### 3.1. Synthesis and Characterization of Materials

Two methods of obtaining polymer-carbon hybrid nanocomposites based on thermostable polydiphenylamine-2-carboxylic acid (PDPAC) [49,50,51] and single-walled carbon nanotubes (SWCNTs) were proposed. Hybrid nanomaterials were synthesized via in situ oxidative polymerization of diphenylamine-2-carboxylic acid (DPAC) in the presence of SWCNTs in the heterophase system in an alkaline medium (pH 11.4) (SWCNT/PDPAC-1) and in an acidic medium (pH 0.3) (SWCNT/PDPAC-2). Meanwhile, in the heterophase system, in an alkaline medium, the monomer and the SWCNTs were in the organic phase (chloroform) and the oxidant (ammonium persulfate) was in an aqueous solution of ammonium hydroxide. For comparison, polymers of diphenylamine-2-carboxylic acid were synthesized under the same conditions. The PDPAC-1 was obtained in NH_4_OH solution in the presence of chloroform. The PDPAC-2 was prepared in 5 M H_2_SO_4_.

Earlier, we showed that during DPAC polymerization in a sulfuric acid solution, the growth of a polymer chain occurs via the C–C bonding into the *para* position of phenyl rings relative to nitrogen [50,51]. The ATR FTIR spectrum of the polymer prepared in 5 M H_2_SO_4_ shows absorption bands at 892 and 803 cm^−1^ related to out-of-plane bending vibrations of δ_C–H_ bonds of 1,4-disubstituted and 1,2,4-trisubstituted benzene rings. The chemical structure of PDPAC-2 obtained in 5 M H_2_SO_4_ (pH 0.3) has the following form:



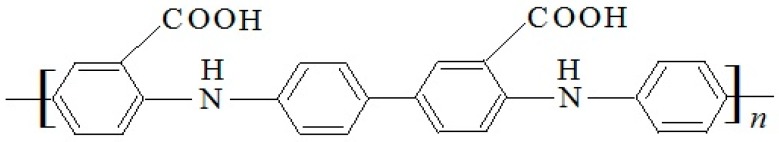



Polymerization of DPAC in the heterophase system in an alkaline medium results in another type of bonding. The ATR FTIR spectrum of the polymer obtained in ammonium hydroxide solution in the presence of chloroform shows the broadening and shift to 753 cm^−1^ of the absorption band at 746 cm^−1^. Also, the ATR FTIR spectrum shows the absence of a band at 892 cm^−1^, and the presence of a wide band at 828 cm^−1^. It all indicates the presence of 1,2-disubstituted and 1,2,4-trisubstituted aromatic rings in the PDPAC-1 structure. This suggests that, during the polymerization of DPAC in an alkaline medium, the growth of a polymer chain proceeds via the C–C bonding in the 2- and 4-positions of phenyl rings relative to nitrogen [49]. The chemical structure of PDPAC-1 prepared in NH_4_OH solution in the presence of chloroform (pH 11.4) has the following form:



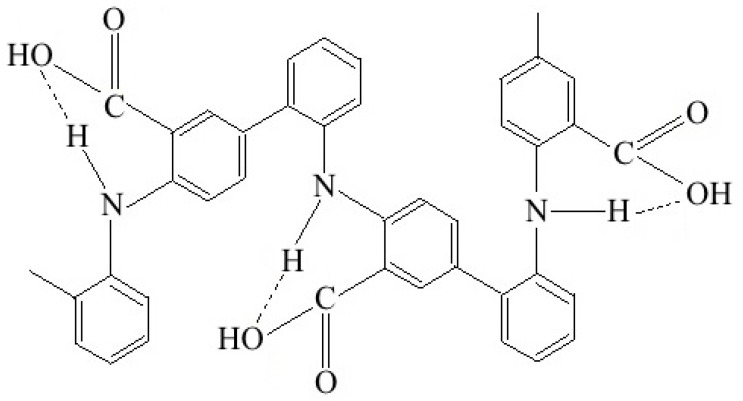



A comparison of ATR FTIR spectra of polymers with nanocomposites prepared under the same conditions was made. It was shown that all the main bands characterizing the chemical structure of PDPAC remain in the SWCNT/PDPAC nanocomposite FTIR spectra. 

Figure 1 shows ATR FTIR spectra of nanocomposites obtained in NH_4_OH solution in the presence of chloroform (SWCNT/PDPAC-1) and in 5 M H_2_SO_4_ (SWCNT/PDPAC-2), depending on the SWCNT concentration. Table 1 shows the assignment of main characteristic absorption bands in the ATR FTIR spectra of PDPAC and SWCNT/PDPAC nanocomposites according to the synthesis method.

Based on the data shown in Figure 1 and Table 1, it can be concluded that the chemical structure of the polymer matrix has a strong dependency on the pH of the reaction medium for the nanocomposite synthesis. It was established that during DPAC polymerization in an acidic medium (pH 0.3), in the presence of SWCNTs, the polymer chain grows via the C–C bonding into the *para* position of phenyl rings relative to nitrogen (δ_C–H_ = 862 and 802 cm^−1^). The process is the same as during the DPAC polymerization under these conditions. In the SWCNT/PDPAC-2 nanocomposite, the absorption bands at 862 and 802 cm^−1^ are due to out-of-plane bending vibrations of δ_C–H_ bonds of 1,4-disubstituted and 1,2,4-trisubstituted benzene rings (Figure 1b).

During the PDPAC/SWCNT-1 nanocomposite synthesis in the heterophase system in an alkaline medium (pH 11.4) in the presence of SWCNTs, the polymer chain grows via the C–C bonding into the 2- and 4-positions of phenyl rings relative to nitrogen (δ_C–H_ = 828 and 748 cm^−1^). The absorption bands at 828 and 748 cm^−1^ in the SWCNT/PDPAC-1 nanocomposite correspond to out-of-plane bending vibrations of the δ_C–H_ bonds of 1,2,4- and 1,2-substituted benzene rings (Figure 1a). The absorption bands at 1680 and 1215 cm^−1^ characterize the stretching vibrations of v_C=O_ in COOH groups. At the same time, COOH groups (v_C=O_ = 1680 and 1215 cm^−1^) are associated with the N–H group (v_N–H_ = 3175 cm^−1^) of the main chain, the same as during the polymerization of DPAC in an alkaline medium [49]. Carboxyl groups along the entire polymer chain form intramolecular hydrogen bonds with amino groups. This is confirmed by the presence in the ATR FTIR spectra of the PDPAC/SWCNT-1 nanocomposite the absorption band at 3264 cm^−1^. This band characterizes the associated COOH---N-H carboxyl groups with the hydrogen bond (Table 1).

The results of the electronic absorption spectra study also confirm the formation of hydrogen bonds in the chain of the polymer matrix in the PDPAC/SWCNT-1 nanocomposite. Figure 2 shows the electronic absorption spectra of SWCNT/PDPAC-1 and SWCNT/PDPAC-2 nanocomposites, depending on the SWCNT concentration. 

As seen in Figure 2a, the maximum at λ*_max_* = 550 nm is present in the electronic spectra of the SWCNT/PDPAC-1 nanocomposite. The peak at λ*_max_* = 550 nm characterizes the electronic transitions of associated COOH—N-H carboxyl groups with the hydrogen bond. This maximum is absent in the electronic absorption spectra of the SWCNT/PDPAC-2 nanocomposite, where carboxyl groups are not associated with amino groups in the structure. At the same time, FTIR spectroscopy data show that carboxyl groups interact with SWCNTs in the acidic medium. ATR FTIR spectra of the SWCNT/PDPAC-2 nanocomposite (Figure 1b), if compared with the polymer PDPAC-2 spectrum, demonstrate a shift of the absorption bands at 1659 and 1226 cm^−1^ to 1655 and 1218 cm^−1^ related to stretching vibrations of v_C=O_ bonds in the COOH groups (Table 1). This shift of the absorption bands by 4 to 8 cm^−1^ indicates the interaction of PDPAC-2 carboxyl groups with the SWCNT surface. This could be caused by the charge transfer through site-selective interaction between the PDPAC-2 carboxyl groups and the SWCNT aromatic structures [36,56].

A characteristic change in the ATR FTIR spectra of the SWCNT/PDPAC-1 nanocomposite compared with the polymer PDPAC-1 spectrum is that the increase in the CNT content results in a hypsochromic shift of the skeletal oscillation frequencies of PDPAC-1 by 10 to 14 cm^−1^ (Figure 1a). The FTIR spectra of the SWCNT/PDPAC-1 nanomaterial demonstrate a shift of the absorption bands at 1595 and 1509 cm^−1^ to 1585 and 1495 cm^−1^, corresponding to the stretching vibrations of v_C–C_ bonds in the aromatic rings (Table 1). This shift indicates the π–π* interaction of PDPAC-1 phenyl rings with the SWCNT aromatic structures. For such materials, the existence of the π–π* interaction between the surface of CNTs and the PANI quinoid units (stacking effect) has been established [23,31,32,36,44]. The formation of polymer on the surface of CNTs provides a charge transfer from the polymer chain to the CNTs [23,36,44]. This is revealed in the shift of the skeletal oscillation frequencies of the polymer.

CP/MAS ^13^C NMR data confirm the growth pattern of the polymer chain proposed above. Figure 3 shows the CP/MAS ^13^C NMR spectra of the PDPAC-1 and PDPAC-2 as well as SWCNT/PDPAC-1 and SWCNT/PDPAC-2 nanocomposites, prepared at C_SWCNT_ = 10 wt %.

The solid-state MAS ^13^C NMR spectrum of the SWCNT/PDPAC-1 nanocomposite retains all signals characterizing the polymer PDPAC-1 (Figure 3a). Both spectra show broad signals from 105 to 155 ppm, with the maximum at δ_C_ = 129 ppm characterizing carbon centers in benzene rings. The signal in the region of δ_C_ = 149 ppm corresponds to carbon atoms of the C–NH groups. The significant broadening of all spectrum signals, especially those around δ_C_ = 129 ppm, is a characteristic change in the CP/MAS ^13^C NMR spectrum of the SWCNT/PDPAC-1 nanocomposite, compared with the PDPAC-1 spectrum. This indicates the interaction of carbon centers in benzene rings with carbon nanotubes, which leads to a decrease in the relaxation time, T_1_, of these centers.

The CP/MAS ^13^C NMR spectrum of the SWCNT/PDPAC-2 nanocomposite, compared with the polymer PDPAC-2 spectrum, shows the increase in signal intensity at 126 and 132 ppm (Figure 3b). This is related to the fact that the paramagnetic centers of CNT reduce the relaxation times, T_1_, of carbon atoms in the polymer. At the same time, previously unobservable quaternary carbon atoms begin to manifest themselves in the NMR spectrum with the increase in the CNT content. The signal at δ_C_ = 146 ppm corresponds to the carbon atoms of the C–NH groups. The wide signal at δ_C_ = 170 ppm characterizes the carboxyl groups.

### 3.2. Morphology of Nanocomposites

The morphology and structure of the obtained hybrid nanomaterials were studied by means of FE-SEM and XRD. Figure 4 shows electron microscopic images of the SWCNT/PDPAC nanocomposites. According to the FE-SEM data, carbon nanotubes are distributed in an amorphous polymer matrix (Figure 4).

Figure 5 shows the diffraction patterns of the SWCNT/PDPAC nanocomposites compared with PDPAC. According to the XRD data, the SWCNT/PDPAC nanocomposites, as well as PDPAC polymers, are amorphous irrespective of the preparing method. The XRD patterns of nanocomposites show an amorphous halo at scattering angles 2θ = 20 to 47°, and a second diffuse halo at 2θ = 60 to 70° characterizing the polymer matrix. The absence of the carbon phase reflection peak in the diffractograms of the nanocomposites is explained by the impossibility of obtaining a diffraction pattern from a single SWCNT plane (Figure 5).

As seen in Figure 4, according to the FE-SEM data, the morphology of the SWCNT/PDPAC nanocomposites depends on the pH of the synthesis reaction medium. As well as of the original PDPAC-1 polymer [49], when the SWCNT/PDPAC-1 nanocomposite is synthesized in the heterophase system, the presence of an organic solvent (chloroform) in an alkaline medium leads to the formation of the polymer matrix morphology with pronounced cavities. These cavities are formed in the places of chloroform drops as the monomer transits from the organic phase to the aqueous one (shown in Figure 4a).

### 3.3. Thermal Properties of Materials 

The thermal stability of the hybrid SWCNT/PDPAC nanocomposites depending on the synthesis method and the SWCNT concentration was studied by the TGA and DSC methods. Figure 6 shows TGA thermograms of the SWCNT/PDPAC nanocomposites compared with PDPAC up to 1000 °C in air and argon flow. The carbon nanotube content in nanocomposites was C_SWCNT_ = 3 and 10 wt % relative to the weight of monomer. Table 2 gives the main thermal characteristics of the materials. 

As can be seen in Figure 6, the weight loss curves of the obtained materials have a stepwise pattern. In this case, the weight loss at low temperatures is associated with the removal of moisture, which is also confirmed by the DSC data. Figure 7 presents DSC thermograms of the SWCNT/PDPAC nanocomposites. An endothermic peak at ~90 to 97 °C is related to the residual moisture removal. The removal of moisture is confirmed by the absence of this endothermic peak on the DSC thermograms of nanocomposites registered after re-heating in an inert atmosphere.

As seen in Figure 6a, both in the PDPAC-1 polymer and in the SWCNT/PDPAC-1 nanocomposites, the weight loss at ~170 °C is connected with the removal of COOH groups [49], despite the fact that they are in the associated COOH—N-H state. In this temperature range, the DSC thermogram has an exothermic peak associated with decomposition. The removal of COOH groups is confirmed by the absence of an exothermic peak at 173 °C on the DSC thermogram plotted after re-heating to 350 °C (Figure 7a). This behavior is also typical of PDPAC-2 obtained in a solution of sulfuric acid [50].

The removal of COOH groups in the SWCNT/PDPAC-1 nanocomposites is also confirmed by FTIR spectroscopy data. Figure 8a presents the ATR FTIR spectra of the 3 wt % SWCNT/PDPAC-1 nanocomposite before and after heating in air at 200 and 300 °C. A comparative analysis of the ATR FTIR spectra of the original nanocomposite and nanocomposites heated to 200 and 300 °C was made. It was shown that the intensity of the bands at 1680 and 1228 cm^−1^, characterizing the COOH groups, decreases as the temperature rises. Additionally, at 300 °C, the band at 1680 cm^−1^ disappears completely. At the same time, the removal of COOH groups begins at temperatures above 150 °C.

As can be seen in Figure 6a, the nanocomposite loses half of the original weight in an inert atmosphere at 834 °C for 3 wt % SWCNT/PDPAC-1, whereas for the PDPAC-1 polymer, this is 663 °C. At 1000 °C, the residue is 44% for 3 wt % SWCNT/PDPAC-1 and 51% for 10 wt % SWCNT/PDPAC-1. The processes of thermo-oxidative degradation of the SWCNT/PDPAC-1 nanocomposites do not depend on the SWCNT content and begin at 350 °C, as well as of the original PDPAC-1 polymer. In air, a 50% weight loss in the SWCNT/PDPAC-1 nanocomposites is observed at 536 to 544 °C. The neat PDPAC-1 polymer loses half of the initial weight in air at 523 °C (Table 2).

As seen in Figure 6b, the absence of weight loss at ~170 °C in the SWCNT/PDPAC-2 nanocomposites is due to the interaction of PDPAC-2 carboxyl groups with the aromatic structures of SWCNTs in the acidic medium. The DSC thermogram presented in Figure 7b does not show thermal effects in this temperature range. The FTIR spectroscopy data confirm the absence of weight loss in the range of 150 to 200 °C. The ATR FTIR spectrum of the 3 wt % SWCNT/PDPAC-2 nanocomposite heated up to 200 °C retains the absorption bands at 1655 and 1218 cm^−1^, which characterize the COOH groups (Figure 8b).

A comparative TGA analysis of the PDPAC-2 polymer and SWCNT/PDPAC-2 nanocomposites obtained at C_SWCNT_ = 3 and 10 wt % was made. It revealed that the main processes of thermal oxidative degradation of nanocomposites begin at 395 and 435 °C, respectively. That is significantly lower than the starting temperature of PDPAC-2 polymer degradation at 570 °C (Table 2). This is due to the fact that in PDPAC-2, after the removal of COOH groups at ~170 °C with the temperature growth, there is the process of additional polymerization, which is induced by atmospheric oxygen [50]. It should be noted that in the argon atmosphere in this temperature range, there is a weight loss in PDPAC-2. In air, a 50% weight loss in the polymer and nanocomposites is observed at 517 °C for PDPAC-2 and 473 °C for 3 wt % SWCNT/PDPAC-2 and 518 °C for 10 wt % SWCNT/PDPAC-2. In an inert atmosphere, PDPAC-2 loses half of its original weight at 396 °C, whereas the SWCNT/PDPAC-2 nanocomposites lose it at 774 to 789 °C. The residue of the SWCNT/PDPAC-2 nanocomposites at 1000 °C is 45% to 46%, which is higher than the same value for PDPAC-2 (35%). 

### 3.4. Electrical Characterization of Materials 

The frequency dependences of the *ac* conductivity (σ*_ac_*) for the PDPAC and the SWCNT/PDPAC nanocomposites were studied. Figure 9 shows the dependence of the conductivity for SWCNT/PDPAC-1 and SWCNT/PDPAC-2 nanocomposites prepared at C_SWCNT_ = 3 and 10 wt % on the *ac* frequency compared with PDPAC-1 and PDPAC-2. Table 3 gives the *ac* conductivity (σ*_ac_*) of materials.

As seen in Figure 9, the PDPAC-1 polymer shows the linear dependence of the conductivity on frequency typical of non-conductive materials [57]. The 3 wt % SWCNT/PDPAC-1 nanocomposite material shows a gradual increase in electrical conductivity over the entire researched frequency range (0.1–10^6^ Hz). As the frequency grows, the electrical conductivity of the 3 wt % SWCNT/PDPAC-1 material increases by four orders of magnitude from 1.93 × 10^−10^ to 1.87 × 10^−6^ S/cm. This nature of frequency dependence of conductivity indicates a hopping mechanism of charge transfer [58,59]. With an increase in the content of SWCNTs from 3 to 10 wt %, the *ac* conductivity increases by six orders of magnitude from 1.93 × 10^−10^ to 2.85 × 10^−4^ Sm/cm (Table 3). In this case, there is a weak frequency dependence of the conductivity at C_SWCNT_ = 10 wt %. This is due to the fact that the 10 wt % SWCNT/PDPAC-1 nanocomposite has passed its percolation threshold.

As for the 10 wt % SWCNT/PDPAC-1, the SWCNT/PDPAC-2 nanocomposites show very weak dependence of the conductivity, σ*_ac_*, on the frequency (Figure 9). As the *ac* frequency grows, the conductivity of the 3 wt % SWCNT/PDPAC-2 nanocomposite increases only from 4.48 × 10^−10^ to 8.07 × 10^−4^ S/cm. However, it should be noted that in the low-frequency range, the conductivity of the 3 wt % SWCNT/PDPAC-2 nanocomposite is significantly higher (by five orders of magnitude) than the conductivity of the 3 wt % SWCNT/PDPAC-1 material. This is due to the fact that during the nanocomposite synthesis in an acidic medium, doping of the polymer component occurs, which makes the main contribution to the 3 wt % SWCNT/PDPAC-2 nanocomposite conductivity. Also, as seen in Figure 9 and Table 3, the *ac* conductivity of the doped PDPAC-2 is much higher than the neutral PDPAC-1 conductivity. The weak frequency dependence of the SWCNT/PDPAC-2 electrical conductivity can be associated with a small value of the imaginary part of the complex dielectric capacitivity, ε”, characteristic of the conductive materials. Therefore, its contribution to conductivity is manifested only at high frequencies [57,60].

Therefore, the results indicate that prepared hybrid thermally stable and electrically conductive nanomaterials have potential applications in the manufacture of supercapacitors [61,62], rechargeable batteries [63,64], sensors [41,45,65,66,67,68], sorbents [69], field-emission devices [70], anti-corrosion coatings [71], dye-sensitized solar cells [72,73,74,75,76,77,78,79], etc.

## 4. Conclusions

Polymer-carbon nanocomposites based on polydiphenylamine-2-carboxylic acid (PDPAC) and single-walled carbon nanotubes (SWCNTs) were synthesized for the first time in the heterophase system in an alkaline medium and in an acidic medium. The dependence of the chemical structure and morphology of the polymer matrix on the pH of the reaction medium of the nanocomposites’ synthesis was shown. It was found that during the polymerization in 5 M H_2_SO_4_ (pH 0.3), the polymer chains grow via the C–C bonding into the *para* position of the phenyl rings relative to nitrogen (δ_C–H_ = 892 and 803 cm^−1^). In the heterophase system in an alkaline medium (pH 11.4), the growth of the polymer chain occurs via the C–C bonding into the 2- and 4-positions of the phenyl rings relative to nitrogen (δ_C–H_ = 828 and 753 cm^−1^). The presence of an organic solvent in the reaction medium leads to a change in the polymer matrix morphology. As a result, cavities formed in the place of chloroform drops. The resulting SWCNT/PDPAC nanocomposites are electrically conductive and thermally stable. The nanocomposites’ *ac* conductivity (σ*_ac_*) relates to the polymer component nature in the hybrid nanomaterial and the CNTs’ presence. In the low-frequency range, because of doping of the polymer component in an acidic medium, the conductivity of the SWCNT/PDPAC-2 nanocomposites is significantly higher than the SWCNT/PDPAC-1 material’s conductivity. It was shown that the SWCNT/PDPAC nanocomposites show weak dependence of the conductivity, σ*_ac_*, on the frequency. In an inert atmosphere, the residue of the SWCNT/PDPAC nanocomposites at 1000 °C was 44% to 51% for SWCNT/PDPAC-1 and 45% to 46% for SWCNT/PDPAC-2.

## Figures and Tables

**Figure 1 polymers-11-01181-f001:**
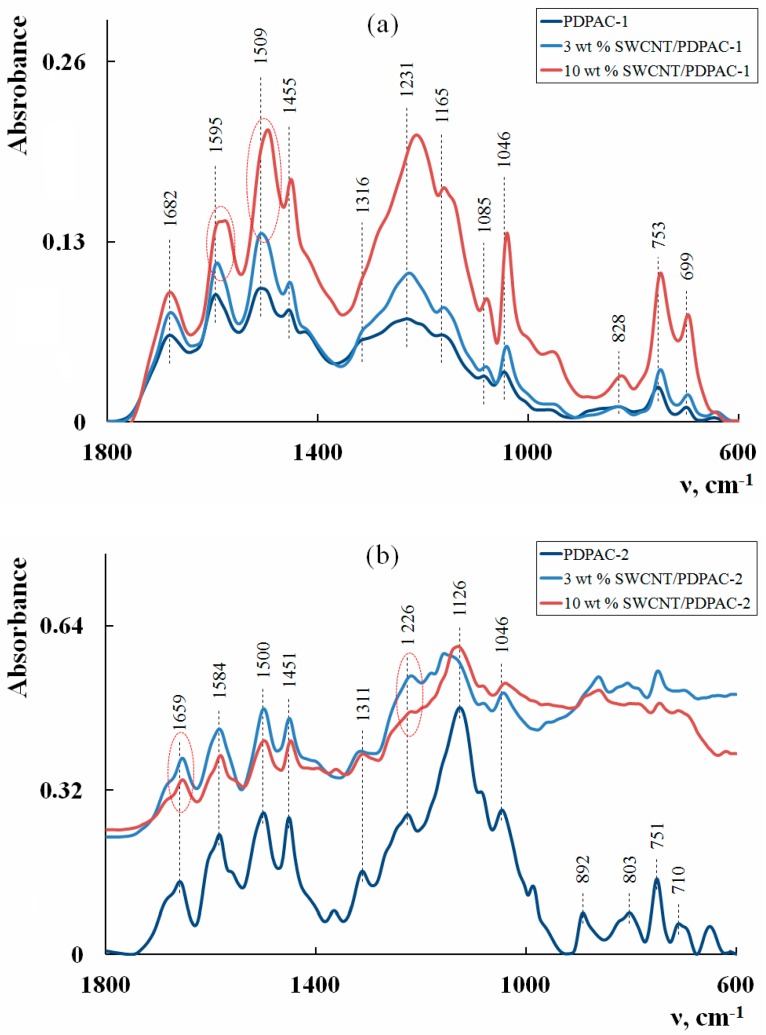
Attenuated total reflection (ATR) FTIR spectra of the PDPAC-1 (**a**) and PDPAC-2 (**b**), and SWCNT/PDPAC-1 (**a**) and SWCNT/PDPAC-2 nanocomposites (**b**).

**Figure 2 polymers-11-01181-f002:**
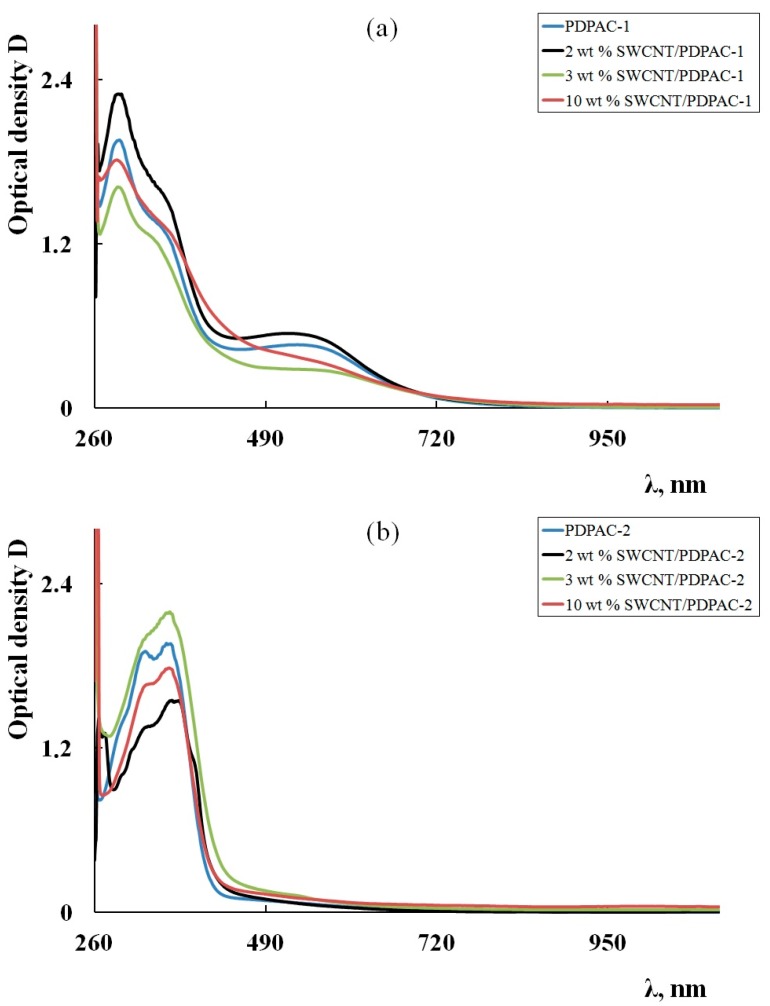
Electronic absorption spectra of the PDPAC-1 (**a**) and PDPAC-2 (**b**), and SWCNT/PDPAC-1 (**a**) and SWCNT/PDPAC-2 nanocomposites (**b**).

**Figure 3 polymers-11-01181-f003:**
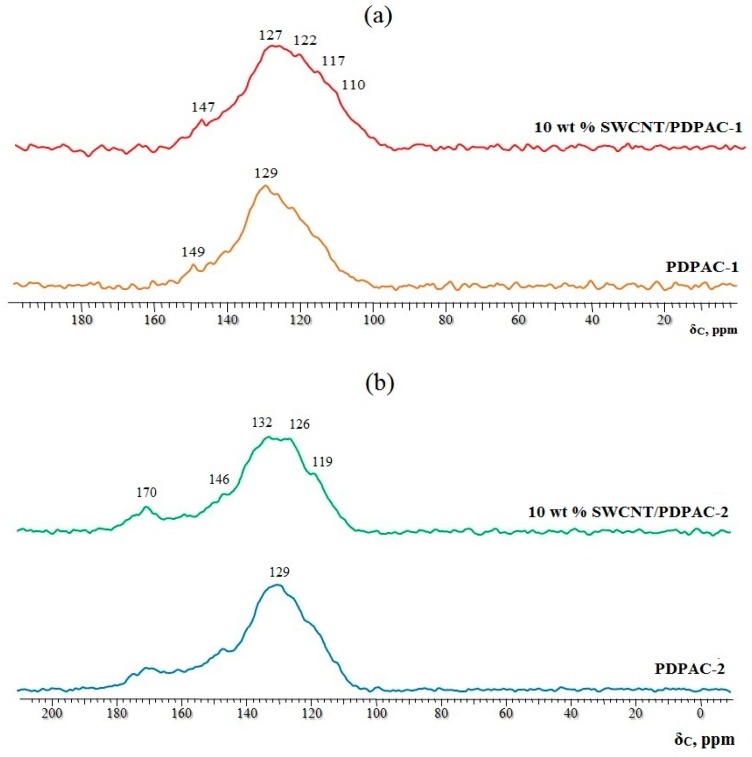
CP/MAS ^13^C NMR spectra of the PDPAC-1 (**a**) and PDPAC-2 (**b**), and SWCNT/PDPAC-1 (**a**) and SWCNT/PDPAC-2 nanocomposites (**b**).

**Figure 4 polymers-11-01181-f004:**
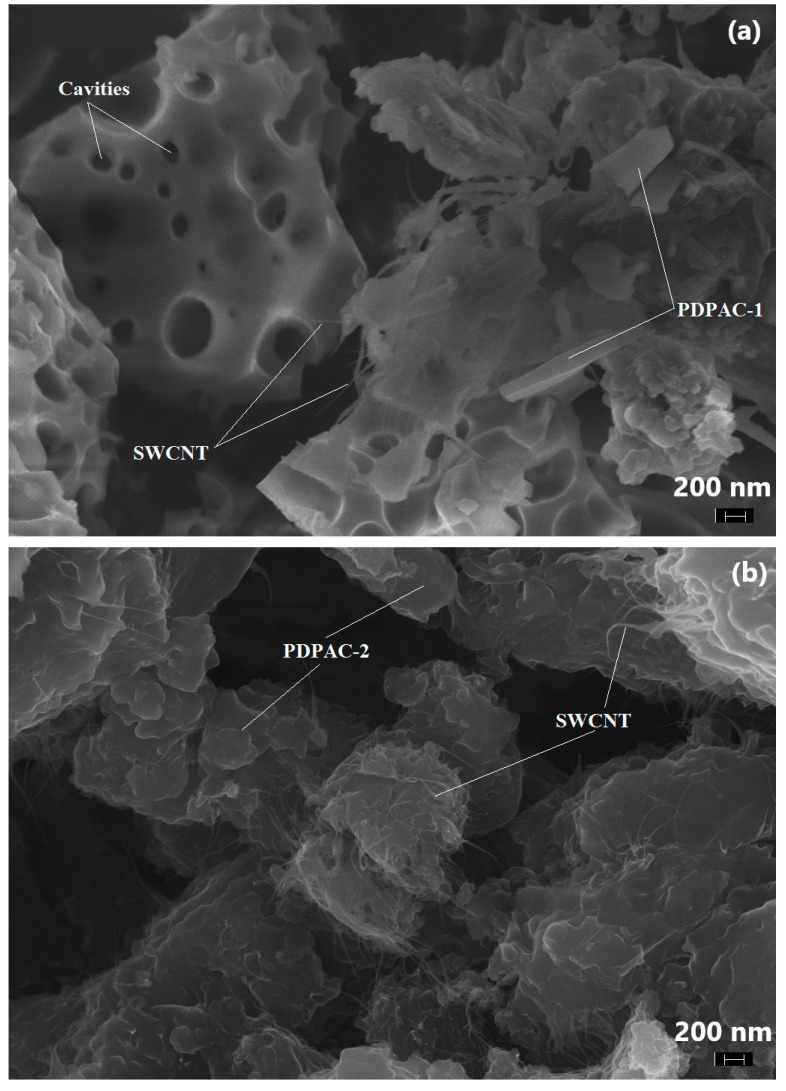
Field emission (FE)-SEM images of the SWCNT/PDPAC-1 (**a**) and SWCNT/PDPAC-2 nanocomposites (**b**).

**Figure 5 polymers-11-01181-f005:**
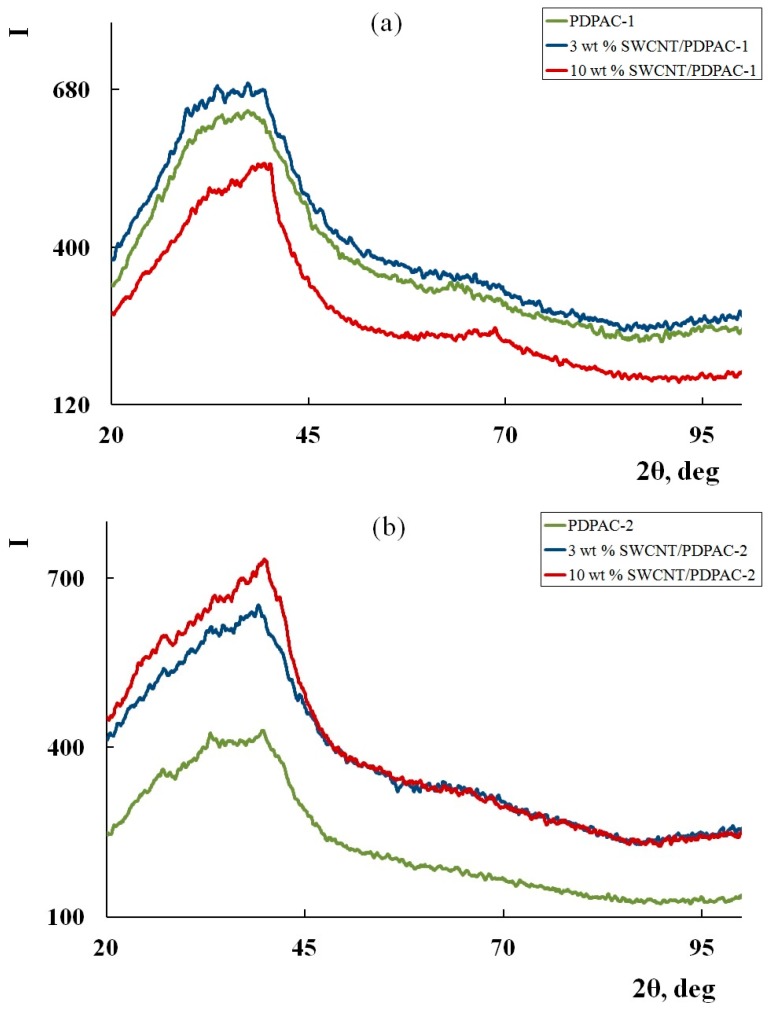
X-ray diffractograms of the PDPAC-1 (**a**) and PDPAC-2 (**b**), and SWCNT/PDPAC-1 (**a**) and SWCNT/PDPAC-2 nanocomposites (**b**).

**Figure 6 polymers-11-01181-f006:**
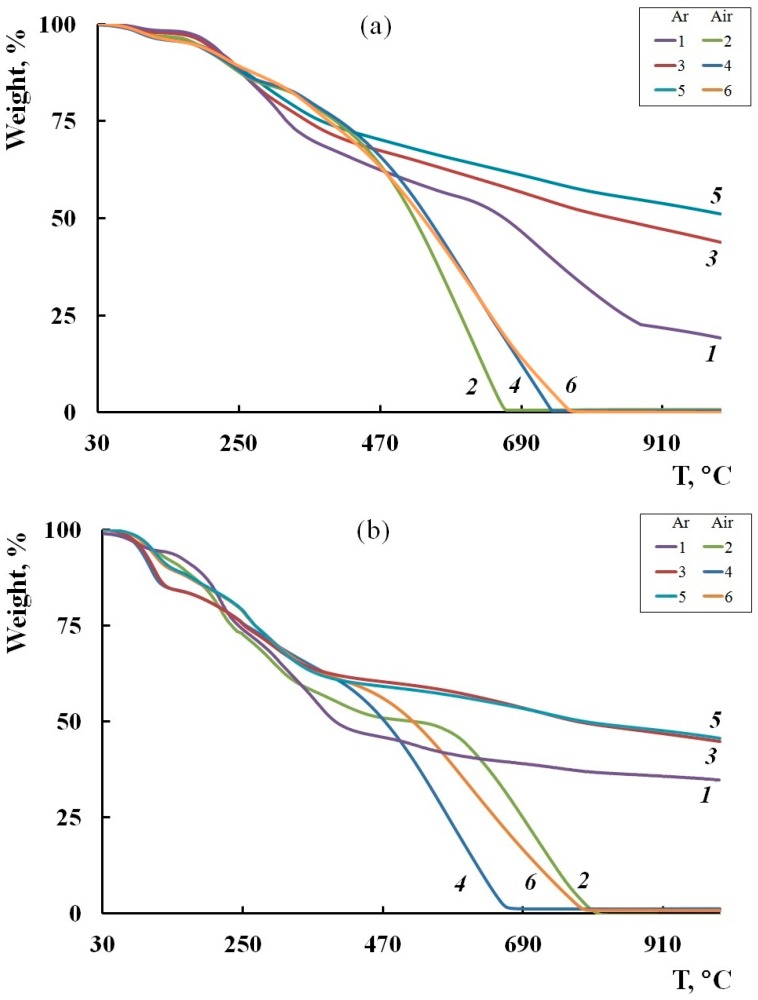
TGA thermograms of the PDPAC-1 (1,2a) and PDPAC-2 (1,2b), and SWCNT/PDPAC-1 (**a**) and SWCNT/PDPAC-2 nanocomposites (**b**), prepared at C_SWCNT_ = 3 (3,4) and 10 wt % (5,6) at heating of up to 1000°C.

**Figure 7 polymers-11-01181-f007:**
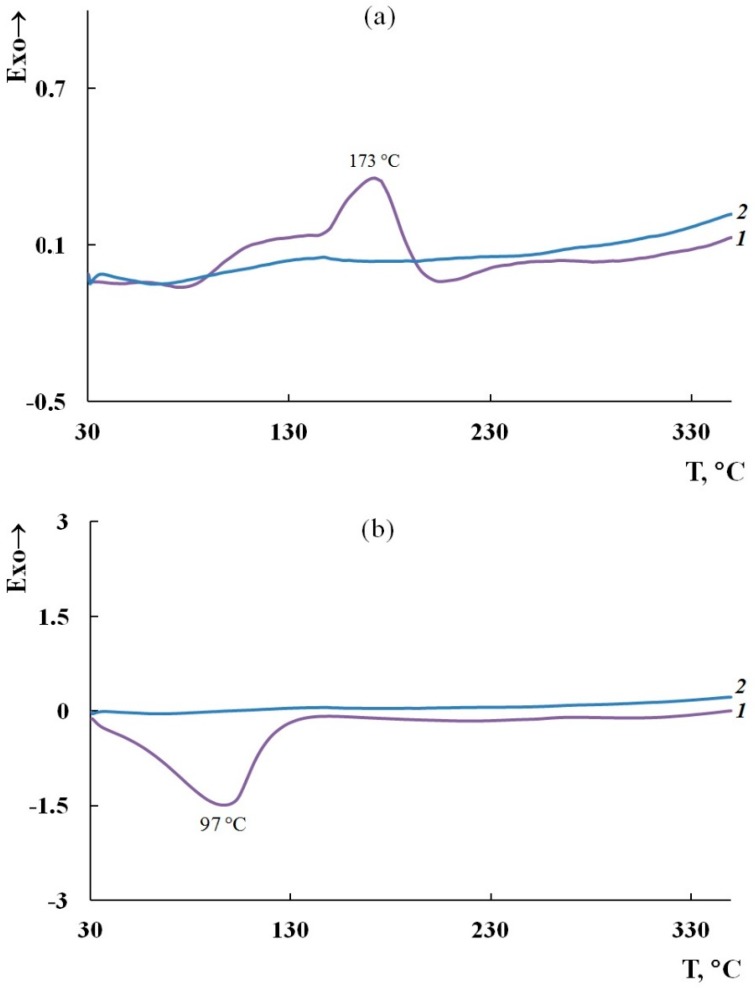
DSC thermograms of the SWCNT/PDPAC-1 (**a**) and SWCNT/PDPAC-2 nanocomposites (**b**) upon heating in the nitrogen flow to 350 °C (1—first heating, 2—second heating).

**Figure 8 polymers-11-01181-f008:**
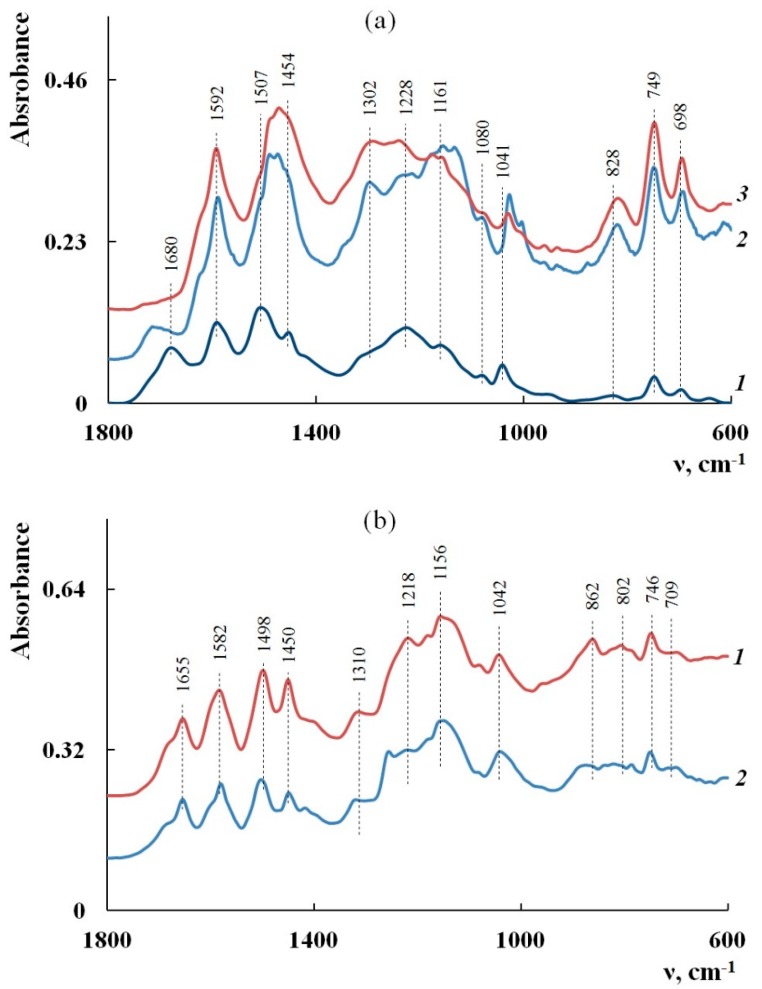
ATR FTIR spectra of the SWCNT/PDPAC-1 (**a**) and SWCNT/PDPAC-2 nanocomposites (**b**) before (1) and after heating in air at 200 (2) and 300 °C (3).

**Figure 9 polymers-11-01181-f009:**
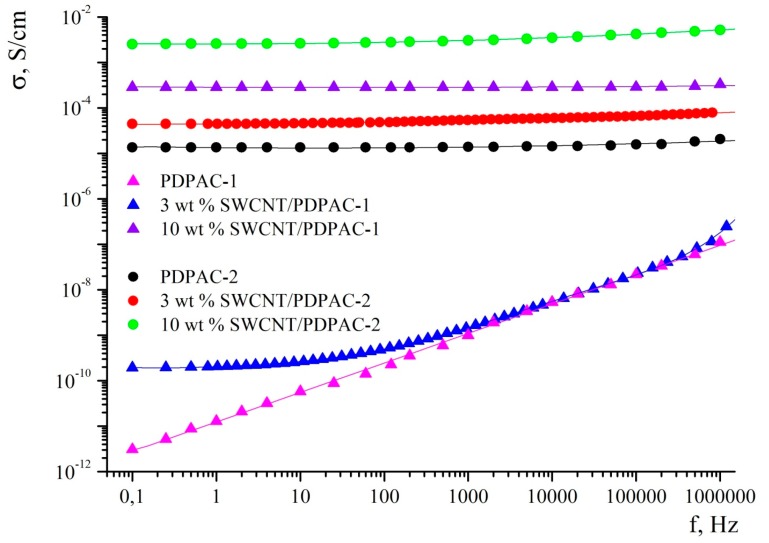
Frequency dependence of the conductivity for the PDPAC and SWCNT/PDPAC nanocomposites.

**Table 1 polymers-11-01181-t001:** Assignment of main characteristic absorption bands in the FTIR spectra of materials.

Assignment of Absorption Bands	Frequency ν, cm^−1^				
	PDPAC-1	SWCNT/PDPAC-1	PDPAC-2	SWCNT/PDPAC-2	DPAC (monomer)
Stretching vibrations of v_N–H_	3236	3175	3303	3308	3337
H—N-H with hydrogen bond	3293	3264	-	-	-
Stretching vibrations of v_C–H_ in an aromatic ring	3081	3062	3020	3033	3034
Stretching vibrations of v_C=O_ in COOH	1682 1231	1680 1215	1659 1226	1655 1218	1658 1259
Stretching vibrations of v_C–C_ in an aromatic ring	1595 1509	1585 1495	1584 1500	1582 1498	1575 1509
Stretching vibrations of v_C–N_	1316	1313	1311	1310	1323
Out-of-plane bending vibrations of δ_C–H_ in an 1,2-substituted aromatic ring	753	748	751	746	746
Out-of-plane bending vibrations of δ_C–H_ in an 1,2,4-trisubstituted aromatic ring	828	828	803	802	-
Out-of-plane bending vibrations of δ_C–H_ in an 1,4-substituted aromatic ring	-	-	892	862	-
Out-of-plane bending vibrations of δ_C–H_ in a mono-substituted aromatic ring	699	696	710	709	697

**Table 2 polymers-11-01181-t002:** Thermal characteristics of materials.

Property	PDPAC-1	SWCNT/PDPAC-1		PDPAC-2	SWCNT/PDPAC-2	
		3 wt %	10 wt %		3 wt %	10 wt %
* *T*_5%_, °C	185/205	174/197	173/178	104/102	88/92	103/107
** *T*_50%_, °C	523/663	544/834	536/>1000	517/396	473/774	518/789
*** Residue, %	20	44	51	35	45	46

* *T*_5%_, ** *T*_50%_—5 and 50% weight losses (air/Ar), *** residue at 1000 °C in the Ar flow.

**Table 3 polymers-11-01181-t003:** The *ac* conductivity (σ*_ac_*) of materials.

Property	PDPAC-1	SWCNT/PDPAC-1		PDPAC-2	SWCNT/PDPAC-2	
		3 wt %	10 wt %		3 wt %	10 wt %
* σ_1_, S/cm	3.10 × 10^−12^	1.93 × 10^−10^	2.85 × 10^−4^	1.35 ×10^−5^	4.48 ×10^−5^	2.52 ×10^−3^
** σ_2_, S/cm	1.12 ×10^−7^	1.87 × 10^−6^	3.32 × 10^−4^	2.17 ×10^−5^	8.07 ×10^−5^	5.15 ×10^−3^

* σ_1_, ** σ_2_—The *ac* conductivity at 0.1 Hz and 1.15 MHz.

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
