# Peer review of "Formation Features of Hybrid Nanocomposites Based on Polydiphenylamine-2-Carboxylic Acid and Single-Walled Carbon Nanotubes"

_polymers, 2019, doi:10.3390/polym11071181_

Round 1
Reviewer 1 Report
Dear authors,
The presented manuscript describes an interesting approach to produce novel PSPAC nanocomposites based on SWCNTs using different conditions and evaluating their main properties.
The introduction of the manuscript partially describes the motivation of the work and the proposed innovations. The authors introduced a lot of references without mentioning them. This part should be reorganized to highlight the novelty of the work and the description of previous work already published.
The experimental part is well organized and described with the most relevant results clearly presented in the manuscript.
In most of the figures the captions are confusing and it is not clear for the reader the assignment of 1, 2 and 3. The caption should mention each figure separately a) PDPAC (1), 3%wt SWCNTS (2) and 10%wt SWCNTs (3) or something similar.
SEM micrographs of PDPAC should be inclued prepared in similar conditions to properly discuss the effect of the SWCNTs in their synthesis.
TEM of neat polymer, raw SWCNTs and 10 wt% should be discussed and presented in the manuscript as it might be relevant for the justification of the conclusions.
In the TGA characterization how you justify the differences in the observed residues of all the studied samples? Are the same trends observed with the 10 wt% composites? please comment on that.
The discussion of TGA results is confusing and the proposed hypothesis needs further discusion and comments.
Finally it would be desirable to present the conductivities of the neat PDPAC and the SWCNT/PDPAC 10 wt% composites.
My recommendation to the authors is to revise the manuscript considering all these comments and to resubmit it before considering it for publication.
With kind regards
Author Response
The authors are grateful to the reviewer for constructive and valuable comments on the manuscript.
The introduction of the manuscript partially describes the motivation of the work and the proposed innovations. The authors introduced a lot of references without mentioning them. This part should be reorganized to highlight the novelty of the work and the description of previous work already published.
We agree with your comment. The introduction part of the manuscript was reorganized to highlight the novelty of the work and the description of previous work already published. Also, the rationale of choosing single-walled nanotubes to form the composites, and the additional contribution of a PDPAC polymer matrix in the development and applications of such nanocomposites was described.
In most of the figures the captions are confusing and it is not clear for the reader the assignment of 1, 2 and 3. The caption should mention each figure separately a) PDPAC (1), 3%wt SWCNTS (2) and 10%wt SWCNTs (3) or something similar.
Appropriate corrections and additions were introduced into the text in colored characters.
SEM micrographs of PDPAC should be inclued prepared in similar conditions to properly discuss the effect of the SWCNTs in their synthesis.
Earlier, SEM micrographs of PDPAC were shown in our work [49].
49. Ozkan, S.Zh.; Eremeev, I.S.; Karpacheva, G.P.; Bondarenko, G.N. Oxidative polymerization of
N-phenylanthranilic acid in the heterophase system. Open J. Polym. Chem. 2013, 3, 63–69.
In this work the morphology of polymers was studied via scanning electron microscopy using electron microscope JSVU3 of JEOE (Japan).
As seen in Figures 5 and 4, the morphology of the PDPAC polymers and the SWCNT/PDPAC nanocomposites is the same and depends on the pH of the synthesis reaction medium. However, the magnifications (and scales), as well as the scanning electron microscopes brands, are different for the SEM micrographs, which makes comparison difficult. Therefore, we have not included these SEM micrographs in our article.
(a)
(b)
Figure 5. SEM images of poly-N-phenylanthranilic acid, obtained by polymerization in the presence of chloroform (a) and in NH4OH solution (b) [49].
Figure 4. Field Emission (FE)-SEM images of the SWCNT/PDPAC-1 (a) and SWCNT/PDPAC-2 nanocomposites (b).
Electron microscopic study was performed by using a Zeiss Supra 25 FE-SEM field emission scanning electron microscope (Carl Zeiss AG, Jena, Germany).
TEM of neat polymer, raw SWCNTs and 10 wt% should be discussed and presented in the manuscript as it might be relevant for the justification of the conclusions.
We fully agree with comments of Reviewer #2 and the TEM images of SWCNT/PDPAC nanocomposites were removed.
In the TGA characterization how you justify the differences in the observed residues of all the studied samples? Are the same trends observed with the 10 wt% composites? please comment on that.
The discussion of TGA results is confusing and the proposed hypothesis needs further discusion and comments.
We agree with your comments. The thermal stability part of the manuscript was reorganized. Also, the thermal stability of the SWCNT/PDPAC nanocomposites, prepared at CSWCNT = 10 wt % was included. Table 2 gives main thermal characteristics of materials. Appropriate corrections and additions were introduced into the text in colored characters.
Finally it would be desirable to present the conductivities of the neat PDPAC and the SWCNT/PDPAC 10 wt% composites.
The frequency-dependent profiles of the neat PDPAC polymers, as well as of the SWCNT/PDPAC nanocomposites, prepared at CSWCNT = 10 wt % were added in the figure 9. Table 3 gives the ac conductivity (sac) of materials. The Electrical Characterization of Materials part of the manuscript was reorganized. Appropriate corrections and additions were introduced into the text in colored characters.

Reviewer 2 Report
The article submitted by Ozkan et al. demonstrates the synthesis and characterizations of single-walled carbon nanotube/polydiphenylamine-2-carboxylic acid (SWCNT/PDPAC) composites. The paper could potentially be accepted in the submitted journal, since the scope of this work is suitable, and the experimental data contain no major flaws; however, some of the discussions are unclear and unreasonable. Additionally, the presentation of this manuscript needs to be improved. Specific comments and questions are below.
1. Please state the rationale of choosing single-walled nanotubes to form the composites.
2. Lines 27-28: The phrase “practical application sphere” is confusing. Consider revising.
3. Please properly segment the “Experimental” section into subsections, e.g., materials synthesis, structural characterizations, electrical characterizations, etc.
4. Table 1: It is unclear why some numbers are bold.
5. Lines 206-207: Please be specific on the “interaction of carboxyl groups with SWCNT”. What is the interaction? How does it shift the associated peaks?
6. Figure 2: The physical meaning of the y-axes is unclear. Additionally, decimal points should be written as “.”, not “,”.
7. Lines 238-239: The “increase in signal intensity from 105 to 125 ppm” is obscure in Figure 3b.
8. Figure 3: The numbers in the figure are confusing. If 1 stands for PDPAC, what does 2 represent in each panel?
9. Lines 264-265: The layered morphology of the composite 2 is invisible in the SEM image.
10. The TEM images of SWCNT/PDPAC-2 are missing.
11. Line 288: The origin of the endothermic peak in the DSC thermogram and the reason of the disappearance of the peak upon re-heating should be explained.
12. Figure 7: The curve “4” shows an abnormal asymptote at a non-zero weight beyond 700 °C. Please justify this observation.
13. Line 268: The conclusion needs one or more references to be backed up.
14. Lines 310-312: The discussions on the thermal stability is unreasonable. The increased weight retention of the composites could be due to the presence of SWCNTs which have intrinsically higher thermal stability than PDPAC. The thermal stability of the polymer itself might not be affected by the SWCNTs. The degradation of the polymer component should be treated as the degradation of the composite.
15. Figure 10: Please add the frequency-dependent profiles of the individual components, i.e., PDPAC and SWCNTs, in the figure.
16. The weak dependence of the electrical conductivity of SWCNT/PDPAC-2 on frequency should be explained.
Author Response
The authors are grateful to the reviewer for constructive and valuable comments on the manuscript.
1. Please state the rationale of choosing single-walled nanotubes to form the composites.
2. Lines 27-28: The phrase “practical application sphere” is confusing. Consider revising.
We agree with your comments. The introduction part of the manuscript was reorganized to highlight the novelty of the work and the description of previous work already published. Also, the rationale of choosing single-walled nanotubes to form the composites, and the additional contribution of a PDPAC polymer matrix in the development and applications of such nanocomposites was described.
3. Please properly segment the “Experimental” section into subsections, e.g., materials synthesis, structural characterizations, electrical characterizations, etc.
The “Experimental” section as well as “Results and Discussion” section was segmented into subsections.
4. Table 1: It is unclear why some numbers are bold.
Appropriate corrections were introduced into the Table 1 in colored characters.
5. Lines 206-207: Please be specific on the “interaction of carboxyl groups with SWCNT”. What is the interaction? How does it shift the associated peaks?
As seen in Figure 2a, the maximum at lmax = 550 nm, which characterizes electronic transitions of associated COOH---N-H carboxyl groups with the hydrogen bond, is present in the electronic spectra of the SWCNT/PDPAC-1 nanocomposite. This maximum at lmax = 550 nm is absent in the electronic absorption spectra of the SWCNT/PDPAC-2 nanocomposite, where carboxyl groups are not associated with amino groups in the structure. At the same time, FTIR spectroscopy data show that carboxyl groups interact with SWCNT in the acidic medium. ATR FTIR spectra of SWCNT/PDPAC-2 nanocomposite (Figure 1b), if compared with the polymer PDPAC-2 spectrum, demonstrate a shift of the absorption bands at 1659 and 1226 cm–1 to 1655 and 1218 cm–1 related to stretching vibrations of nC=О bonds in COOH groups (Table 1). This shift of the absorption bands by 4–8 cm–1 indicates the interaction of PDPAC-2 carboxyl groups with the SWCNT surface. This could be caused by the charge transfer through site-selective interaction between the PDPAC-2 carboxyl groups and the SWCNT aromatic structures [36, 56]. Appropriate corrections and additions were introduced into the text in colored characters.
36. Cochet, M.; Maser, W.K.; Benito, A.M.; Callejas, M.A.; Martínez, M.T.; Benoit, J.-M.; Schreiber, J.; Chauvet, O.
Synthesis of a new polyaniline/nanotube composite: “in-situ” polymerization and charge transfer through
site-selective interaction. Chem. Commun. 2001, 37, 1450–1451.
56. Peng, H.; Alemany, L.B.; Margrave, J.L.; Khabashesku, V.N. Sidewall carboxylic acid functionalization of
single-walled carbon nanotubes. J. Am. Chem. Soc. 2003, 125, 15174–15182.
6. Figure 2: The physical meaning of the y-axes is unclear. Additionally, decimal points should be written as “.”, not “,”.
7. Lines 238-239: The “increase in signal intensity from 105 to 125 ppm” is obscure in Figure 3b.
8. Figure 3: The numbers in the figure are confusing. If 1 stands for PDPAC, what does 2 represent in each panel?
Appropriate corrections and additions were introduced into the text in colored characters.
9. Lines 264-265: The layered morphology of the composite 2 is invisible in the SEM image.
Appropriate corrections were introduced into the text.
10. The TEM images of SWCNT/PDPAC-2 are missing.
We fully agree with comments of Reviewer #2 and the TEM images of SWCNT/PDPAC nanocomposites were removed.
11. Line 288: The origin of the endothermic peak in the DSC thermogram and the reason of the disappearance of the peak upon re-heating should be explained.
The weight loss in the TGA thermograms of the SWCNT/PDPAC nanocomposites at low temperatures is associated with the removal of moisture, which is also confirmed by the DSC data. Figure 7 presents DSC thermograms of the SWCNT/PDPAC nanocomposites. An endothermic peak at ~ 90–97 °С is related to the residual moisture removal. The removal of moisture is confirmed by the absence of this endothermic peak on the DSC thermograms of nanocomposites registered after re-heating in an inert atmosphere. Appropriate corrections and additions were introduced into the text in colored characters.
12. Figure 7: The curve “4” shows an abnormal asymptote at a non-zero weight beyond 700 °C. Please justify this observation.
13. Line 268: The conclusion needs one or more references to be backed up.
14. Lines 310-312: The discussions on the thermal stability is unreasonable. The increased weight retention of the composites could be due to the presence of SWCNTs which have intrinsically higher thermal stability than PDPAC. The thermal stability of the polymer itself might not be affected by the SWCNTs. The degradation of the polymer component should be treated as the degradation of the composite.
We agree with your comments. The thermal stability part of the manuscript was reorganized. Also, the thermal stability of the SWCNT/PDPAC nanocomposites, prepared at CSWCNT = 10 wt % was included. Table 2 gives main thermal characteristics of materials. The conclusion was backed up by one reference. Appropriate corrections and additions were introduced into the text in colored characters.
15. Figure 10: Please add the frequency-dependent profiles of the individual components, i.e., PDPAC and SWCNTs, in the figure.
16. The weak dependence of the electrical conductivity of SWCNT/PDPAC-2 on frequency should be explained.
The frequency-dependent profiles of the PDPAC polymers, as well as of the SWCNT/PDPAC nanocomposites, prepared at CSWCNT = 10 wt % were added in the figure 9. Table 3 gives the ac conductivity (sac) of materials. The Electrical Characterization of Materials part of the manuscript was reorganized.
The SWCNT/PDPAC-2 nanocomposites show very weak dependence of the conductivity sac on frequency (Figure 9). As the ac frequency grows, the conductivity of the 3 wt % SWCNT/PDPAC-2 nanocomposite increases only from 4.48 × 10–5 S/cm to 8.07 × 10–5 S/cm. However, it should be noted that in the low-frequency range the conductivity of the 3 wt % SWCNT/PDPAC-2 nanocomposite is significantly higher (by 5 orders of magnitude) than the conductivity of the 3 wt % SWCNT/PDPAC-1 material. This is due to the fact that during the nanocomposite synthesis in an acidic medium doping of the polymer component occurs, that makes the main contribution to the 3 wt % SWCNT/PDPAC-2 nanocomposite conductivity. Also, as seen in Figure 9 and Table 3, the ac conductivity of doped PDPAC-2 is much higher than the neutral PDPAC-1 conductivity. The weak frequency dependence of the SWCNT/PDPAC-2 electrical conductivity can be associated with a small value of the imaginary part of the complex dielectric capacitivity ε", characteristic of the conductive materials. Therefore, its contribution to conductivity is manifested only at high frequencies [57, 60].
57. Eletskii, A.V.; Knizhnik, A.A.; Potapkin, B.V.; Kenny, J.M. Electrical characteristics of carbon nanotube
doped composites. Uspekhi Phyzicheskikh Nauk. 2015, 185, 225–270.
60. Coleman, J.N.; Curran, S.; Dalton, A.B.; Davey, A.P.; McCarthy, B.; Blau, W.; Barklie R.C.
Percolation-dominated conductivity in a conjugated-polymer-carbon-nanotube composite. Phys. Rev. B.
1998, 58, R7492(R)– R7495(R).
Appropriate corrections and additions were introduced into the text in colored characters.
Also, English was significantly improved.

Reviewer 3 Report
Reviewer’s comments on the manuscript
Formation of features of hybrid nanocomposites based on polydiphenylamine-2-carboxylic acid and single-walled carbon nanotubes
This paper deals with the elaboration, through two different routes, of nanocomposites constituted of a matrix PDPAC (used for the first time) filled with single-walled carbon nanotubes. A comparative study between the two materials is carried out. This interesting paper is well-written.
Major remarks:
What is the additional contribution of a PDPAC polymer matrix in the development and applications of such nanocomposites?
Could the authors clarify what "synergistic effect" means in line 27?
Should not Fig. 1a have the same ordinate scale as Fig. 1b in order to compare both systems? Likewise, the abscissa scale should be identical in Fig. 2a and 2b. What does D mean on the y-axis?
For this kind of materials, the preparation of ultrathin sections of few tens of nanometers for TEM observations usually requires the use of an ultracryomicrotome. I am not sure that a grinding allows a correct observation of the samples. In this context, the TEM micrographs, shown in Figure 5, seem unconvincing to me. Authors should choose more representative TEM micrographs or remove them.
In the same way, the magnifications (and scales) are different for the two SEM micrographs presented in Fig.4 for nanocomposites based on PDPAC-1 and PDPAC-2, which makes comparison difficult.
The authors should also clearly show the structural entities in the figures.
As far as XRD results are concerned, shoulders around 60° for some materials and a diffraction peak at ~ 70° for the 10%SWCNT/PDPAC-1 nanocomposite are not discussed. Why?
Minor remarks:
In the "experimental” section, it would be relevant to separate the "materials" part from the "characterizations" part.
The paragraph (143-146) states that the authors have elaborated nanocomposites, but the results have not yet been shown. This paragraph should be deleted or moved after the results.
I think that the absorption band of the SWCNT/PDPAC-1 is 748 cm-1 (line 182 and Cf. Table 1) and not 753 cm-1.
The order of acronyms (“XRD, TEM and FE-SEM” line 255) should follow the same order as the presentation of the results in the text.
In the caption of Fig. 10, replace (a) and (b) by (1) and (2)
Considering the above-points, I do not recommend acceptance of the paper. Major revisions, concerning microscopic and XRD characterizations more particularly, are expected. The interest of the polymer studied is to be specified.

Author Response
The authors are grateful to the reviewer for constructive and valuable comments on the manuscript.
What is the additional contribution of a PDPAC polymer matrix in the development and applications of such nanocomposites?
Could the authors clarify what "synergistic effect" means in line 27?
We agree with your comments. The introduction part of the manuscript was reorganized to highlight the novelty of the work and the description of previous work already published. Also, the rationale of choosing single-walled nanotubes to form the composites, and the additional contribution of a PDPAC polymer matrix in the development and applications of such nanocomposites was described.
Should not Fig. 1a have the same ordinate scale as Fig. 1b in order to compare both systems? Likewise, the abscissa scale should be identical in Fig. 2a and 2b. What does D mean on the y-axis?
Appropriate corrections and additions were introduced into the text.
For this kind of materials, the preparation of ultrathin sections of few tens of nanometers for TEM observations usually requires the use of an ultracryomicrotome. I am not sure that a grinding allows a correct observation of the samples. In this context, the TEM micrographs, shown in Figure 5, seem unconvincing to me. Authors should choose more representative TEM micrographs or remove them.
We fully agree with your comments and the TEM images of SWCNT/PDPAC nanocomposites were removed.
In the same way, the magnifications (and scales) are different for the two SEM micrographs presented in Fig. 4 for nanocomposites based on PDPAC-1 and PDPAC-2, which makes comparison difficult.
The authors should also clearly show the structural entities in the figures.
Appropriate corrections of the SEM micrographs were made.
As far as XRD results are concerned, shoulders around 60° for some materials and a diffraction peak at ~ 70° for the 10%SWCNT/PDPAC-1 nanocomposite are not discussed. Why?
According to the XRD data, the SWCNT/PDPAC nanocomposites, as well as PDPAC polymers are amorphous irrespective of the preparing method. The XRD patterns of nanocomposites show an amorphous halo at scattering angles 2θ = 20–47°, and a second diffuse halo at 2q = 60–70° characterizing the polymer matrix. Appropriate corrections and additions were introduced into the text in colored characters.
In the "experimental” section, it would be relevant to separate the "materials" part from the "characterizations" part.
The “Experimental” section as well as “Results and Discussion” section was segmented into subsections.
The paragraph (143-146) states that the authors have elaborated nanocomposites, but the results have not yet been shown. This paragraph should be deleted or moved after the results.
We agree with your comment. This paragraph was deleted.
I think that the absorption band of the SWCNT/PDPAC-1 is 748 cm-1 (line 182 and Cf. Table 1) and not 753 cm-1.
We fully agree with your comment and have made appropriate changes.
The order of acronyms (“XRD, TEM and FE-SEM” line 255) should follow the same order as the presentation of the results in the text.
In the caption of Fig. 10, replace (a) and (b) by (1) and (2).
Appropriate corrections and additions were introduced into the text in colored characters.

Round 2
Reviewer 2 Report
The authors have properly addressed all the comments raised by the reviewers. The revised manuscript is qualified to be published.
Author Response
The authors are grateful to the reviewer for the recommendation to publish this article.

Reviewer 3 Report
Reviewer’s comments on the manuscript
Formation of features of hybrid nanocomposites based on polydiphenylamine-2-carboxylic acid and single-walled carbon nanotubes
This paper deals with the elaboration, through two different routes, of nanocomposites constituted of a matrix PDPAC (used for the first time) filled with single-walled carbon nanotubes. A comparative study between the two materials is carried out. The authors took into account the reviewers' numerous comments and the corrected version was greatly improved although some sentences are still too long: for example the one reported in the abstract (Polymer-carbon hybrid … ammonium hydroxide)
So, considering the improvement of the corrected paper, I recommend its acceptance, subject to minor changes to be made to some too long sentences.

Author Response
The authors are grateful to the reviewer for the recommendation to publish this article.
We agree with your comments. The manuscript was revised according to comments. Some too long sentences were reorganized. Appropriate corrections were introduced into the text in colored characters.
